# Superior Adaptations in Adolescent Runners Using Heart Rate Variability (HRV)-Guided Training at Altitude

**DOI:** 10.3390/bios11030077

**Published:** 2021-03-11

**Authors:** Petr Bahenský, Gregory J. Grosicki

**Affiliations:** 1Department of Sports Studies, Faculty of Education, University of South Bohemia, 371 15 České Budějovice, Czech Republic; 2Biodynamics and Human Performance Center, Georgia Southern University, Savannah, GA 31419, USA; ggrosicki@georgiasouthern.edu

**Keywords:** heart rate variability, altitude training, cardiac autonomic, youth, runners, coaching

## Abstract

We evaluated the efficacy of heart rate variability (HRV)-guided training in adolescent athletes during a 2-week, high altitude (≈1900 m) training camp. Sixteen middle- and long-distance runners (4 female/12 male, 16.9 ± 1.0 years, 65.44 ± 4.03 mL·kg^−1^·min^−1^) were divided into 2 matched groups, both of which received the same training plan, but one of which acquired postwaking HRV values that were used to tailor the training prescription. During the camp, seven athletes in the HRV-guided group combined for a total of 32 training adjustments, whereas there were only 3 runners combined for 14 total training adjustments in the control group. A significant group by time interaction (*p* < 0.001) for VO_2_max was driven by VO_2_max improvements in the HRV group (+2.8 mL·kg^−1^·min^−1^, +4.27%; *p*_Bonf_ = 0.002) that were not observed in the control condition (+0.8 mL·kg^−1^·min^−1^, +1.26%; *p*_Bonf_ = 0.643). After returning from the camp, all athletes in the HRV group set a personal best, and six out of eight achieved their best positions in the National Championship, whereas only 75% of athletes in the control group set a personal best and five out of eight achieved their best positions in the National Championship. These data provide evidence in support of HRV-guided training as a way to optimize training prescriptions in adolescent athletes.

## 1. Introduction

Managing a training load is an essential process in all sports. In high-end competition, athletes often find themselves balancing a fine line between success and overtraining [1,2,3,4]. Moreover, even elite athletes may respond differently to the same stimulus [5]. In endurance athletes, subjective and objective parameters may be used to track a training load, thus helping to individualize exercise prescriptions. Individualization of a training load is an essential element of athletic success [1,3,5] that may help to prevent injury and overtraining [6,7,8].

Heart rate variability (HRV)-guided exercise training has been studied for more than a decade [9,10] and has become increasingly popular in recent years due to advances in wearable technology that have increased the accessibility of HRV metrics [11,12,13,14,15,16]. Rapid, reliable, and sensitive HRV metrics permit noninvasive evaluations of autonomic homeostasis (parasympathetic and sympathetic activity), with a higher HRV value reflecting parasympathetic dominance and physical activity readiness [9,17]. While several HRV-guided training strategies have been proposed [18], recent work using individual HRV values to determine a compensation level to manage the training process has proven to be particularly effective [19,20].

Previous studies have provided evidence in support of HRV-guided training as a means to promote superior adaptations when compared to conventional training means. However, due to the complexity of monitoring HRV data and manipulating the training load, these data often come from case studies [21]. In a relatively modest group of well-trained cyclists (*n* = 20), evidence for the superiority of HRV-guided vs. block-periodization training was recently provided by Javaloyes and colleagues [22], though similar improvements in VO_2_max between the HRV-guided and traditional training in well-trained runners have also been observed [23]. Though limited in number, some studies in larger samples have shown that HRV-guided training may help to prevent the development of overtraining [23,24], which may prove particularly useful at altitude, where the risk of overtraining is increased [25,26,27]. In young athletes, the risk for overtraining is exacerbated by dynamic and rapidly changing physiological profiles [28,29,30,31], yet few studies have sought to evaluate the efficacy of HRV-guided training in this population [11].

The overarching goal of the present study was to evaluate the efficacy of HRV-guided training in adolescent runners during a short-duration, high-altitude training camp. We hypothesized that HRV-guided training would promote superior physiological adaptations (i.e., an increased VO_2_max), which is reflected by personal best performances in each athlete’s respective discipline. Furthermore, we anticipated that athletes participating in HRV-guided training would make more training adjustments when compared to a control group that followed the same predetermined training prescription.

## 2. Materials and Methods

### 2.1. Subjects

The participants were 16 middle- and long-distance adolescent runners (16.9 ± 1.0 years, 66.4 ± 6.0 kg, 179.1 ± 7.2 cm, 65.44 ± 4.03 mL·kg^−1^·min^−1^), all of which were finalists for the National Championship in the Czech Republic. This convenience sample was comparable to previous works in the field that similarly featured between 10–20 participants [22,31,32,33,34,35,36]. The runners were matched for gender, height, weight, and cardiorespiratory fitness, and allocated into one of two groups: an HRV-guided training group and a control group using a matched-pairs study design [37]. Prior to the study, all runners trained under the same coach for approximately the same duration (2.1 ± 1.1 years) and provided written consent to participate in the study following an explanation of the risks and benefits. The study protocol was approved by the local Ethics Committee (no. 001/2018) and followed the guidelines laid down by the World Medical Assembly Declaration of Helsinki.

### 2.2. Study Design

The runners took part in a 2-week high-altitude training camp (location) in three different zones:-1000 m above sea level—training at a stadium.-1600 m above sea level—training around a lake.-1900 m above sea level—in the vicinity of their accommodation.

In consideration of the physiological implications of altitude exposure, the training intensity for the first five days was reduced by 20 s·km^−1^ at 1900 m, by 15 s·km^−1^ at 1600 m, and by 10 s·km^−1^ at 1000 m compared to normal training paces for the athletes in their usual place of residence (≈400 m above sea level). After the initial acclimatization period, the speed was reduced by ≈5 s·km^−1^. We decided to modify the training plan based on the external, rather than internal load, as a prior training prescription had been performed using pace goals (i.e., external load). The basic structure of the training plan was identical in both groups and is provided in Table 1.

The experimental group began postwaking HRV measurements 4 weeks before the training camp to acquire baseline values and continued to acquire HRV throughout the camp. Immediately after waking and bladder voiding, athletes applied a chest strap (mySASY, Olomouc, Czech Republic) that was used to interface with a Bluetooth smartphone application (mySASY mobile). The HRV measurement protocol took approximately 15 min and consisted of an initial 120 heartbeats in the supine position, 360 heartbeats in the standing position, and then 360 heartbeats in the supine position. R-R intervals (the intervals between successive heartbeats) from this orthoclinostatic test [38] were processed using spectral HRV analysis to generate an HRV total score (TS), as described elsewhere [39]. Daily TS’s were used to generate an HRV compensation value on a scale of 50–150%, where a lower value indicated reduced HRV (sympathetic dominance) and a higher value indicated greater cardiac–vagal modulation (parasympathetic dominance). Daily HRV compensation values were computed in the context of the average (AVG) and standard deviation (SD) of the TSs from the past 7 to 20 days using the following interval limits: min = 50|AVG − SD × 1.5 = 75|AVG − SD = 90|AVG + SD = 110|AVG + SD × 1.5 = 130|max = 150. The daily TSs were inversely interpolated to the resulting interval range and then the HRV compensation value was subsequently derived.

The HRV compensation scores were then fit to a predetermined algorithm that was recommended by mySASY [40,41] (Figure 1), which was used to monitor and potentially adjust the training load. Briefly, if the compensation value was <85%, the athlete did not subsequently complete quality training (i.e., speed, intervals, etc.), even if they had it planned. If the compensation value was >115%, quality training was included, even if it was not planned, but for a maximum of 2 days in a row. With compensation values between 85 and 115%, standard, moderate-intensity training was used. The measured HRV compensation value was then used to determine each athlete’s readiness for each day of training [40], with the main training phase taking place each morning, roughly 3–4 h after the HRV measurement. In the control group, the adjustments to training were only made on the basis of subjective strain, as evaluated by each athlete. All training adjustments were tracked by the research team.

Maximal graded exercise testing on a treadmill (Lode Valiant 2 Sport, Lode B.V., Groningen, The Netherlands) was performed by all athletes 3 days before attending and 8 days after returning from the 2-week training camp. Athletes were instructed to avoid alcohol and caffeinated beverages 24 h prior to the test and to avoid intensive training 48 h prior. All athletes ate a meal ≈3 h prior to the test, which was performed during the same time of day for chronobiologic control. The test began with a 4 min warm-up at an initial speed of 6 km·h^−1^, after which the speed was subsequently increased to 12 and 9 km·h^−1^ for the boys and girls, respectively. The speed was subsequently increased by 1 km·h^−1^ every minute until volitional exhaustion. The total test time for all athletes was 7–9 min and the achievement of VO_2_max was verified by all athletes meeting the following criteria: respiratory exchange ratio (RER) >1.1, rating of perceived exertion (RPE) >17, achievement of a plateau in VO_2_, and achievement of a 90% age-predicted maximal heart rate. Following their return from the altitude training camp, the race performances of all athletes were also tracked for 8 weeks, and the achievements of personal best times and National Championship performances were also noted as additional variables for comparison.

### 2.3. Statistical Analysis

Data are presented as mean ± SD. The normality of data was confirmed using a the Shapiro–Wilk test and boxplots. A two-way repeated-measures ANOVA (group × time) was used to compare changes in VO_2_max before and after the training camp in the HRV-guided and control training groups and significant interactions were examined using Bonferroni adjusted simple main effect post hoc comparisons. The data processing was done in Excel 2016 (Microsoft, Oregon, WA, USA) and Statistica (StatSoft, Tulsa, OK, USA).

## 3. Results

The HRV compensation values are provided in Figure 2. As anticipated, the HRV compensation values declined during the first 72 h of altitude exposure, followed by some minor oscillations over the next week. Interestingly, around day 9, the HRV compensation values appeared to rise, and this trend continued for the next five days. Over the 2-week training camp, seven runners in the HRV guided training group adjusted their training, accounting for a total of 32 training adjustments. Meanwhile, in the control group, only 4 runners adjusted their training (14 total adjustments).

The VO_2_max values measured 3 days before and 8 days after the training camp in the HRV-guided training group and the control group are provided in Table 2. A significant group × time interaction (*p* < 0.001) was driven by significant improvements in VO_2_max in the HRV-guided training group (+4.27%; *p*_Bonf_ = 0.002) that were not observed in the control group (+1.26%; *p*_Bonf_ = 0.643).

Table 3 shows the descriptive statistics regarding the number of athletes setting personal records and achieving their personal best positions in the Czech National Championship within 8 weeks of their return to sea level. In the HRV-guided training group, all athletes set personal records and six out of eight (75%) achieved their best position in the National Championship. Meanwhile, in the control group, six out of eight athletes set personal records and five out of eight achieved their best position in the National Championship. These data are consistent with the measured VO_2_max values.

## 4. Discussion

In adolescent runners participating in a 2-week altitude training camp, we observed superior adaptations in cardiorespiratory fitness (VO_2_max) with the HRV-guided compared to conventional training methods. Relevantly, superior aerobic benefits in the HRV-guided training group appeared to translate to meaningful improvements in race performance, as evinced by a greater number of personal records and successful performances in the National Championship. These data add to a growing body of literature providing evidence in support of the efficacy of HRV-guided training as a means to optimize training adaptations and extend upon these works by highlighting the utility of HRV-guided training as a useful tool in adolescent athletes participating in training under extreme conditions (i.e., altitude exposure).

Due to the unique physiological stress that is inflicted by altitude exposure, it is traditionally recommended that athletes initially reduce their training volume to allow for acclimation [42]. Furthermore, growth and development processes associated with adolescence may increase nutritional and sleep needs in a manner that nuances the recovery process. As such, young individuals training at altitude may be considered particularly at risk for overtraining. Not only did we observe robust and homogenous benefits in all athletes using the HRV-guided training program but no injuries, sicknesses, or other signs of overtraining were reported in this group. This is in line with data in older individuals under normal environmental conditions [6,10,43,44] and suggests that HRV-guided training may not only help to optimize training efficiency but also may help to reduce the risk of overtraining and injury [24]. It is also important to note that while the training loads were manipulated differentially between subjects participating in the HRV-guided training group, and in one subject, no training adjustments were made at all. This observation highlights how differently even athletes of a similar skill level may handle a fixed training stimulus and emphasizes the value of individualized training paradigms [1,3,5,45]. Meanwhile, the members of the control group achieved standard changes in performance [32].

A novel aspect of the present study was our use of frequency-domain HRV parameters to inform the training recommendations. Conversely, seminal work by Vesterinen and colleagues used the time-domain HRV parameter RMSSD (Root mean square of successive R-R interval differences) [23], where this approach has been repeated by many other HRV guided training studies [22,33]. Indeed, time-domain HRV parameters may be preferable when attempting to capture short-duration HRV measures [46]. However, because we were specifically interested in the use of an orthoclinostatic test [47] to possibly provide more detailed insight into autonomic statuses, we found longer-duration (i.e., 5 min), frequency-based HRV acquisition to be more appropriate. Future studies comparing short-duration time-domain HRV indices (i.e., RMSSD) to more time-intensive cardiac autonomic responses to the shifting of body positions captured using HRV frequency-domain parameters as a means to evaluate physical activity readiness are of interest.

Increases in the HRV and reductions in the HR values throughout an altitude training cycle have previously been observed [48,49,50] and are thought to represent acclimatization to the environmental stimulus. In the present study, we similarly observed an upward trend (via visual inspection) in HRV values, beginning on day 9 of the training camp (Figure 2), providing evidence that the training was well-tolerated. A strength of analyzing trends in these data, rather than performing formal hypothesis testing, is that the former is likely more reflective of the way a coach or trainer working with these athletes would interact with these values. It would have been interesting to document changes in the HRV in the control group to determine whether a similar pattern was observed, or whether the HRV values might have continued to trend downward throughout the training camp. Other limitations of this work include our relatively small number of participants, as well as the fact that the groups were at camp together, and thus knowledge of the group randomization might have influenced the athletes. Finally, it should be kept in mind that no formal hypothesis testing was conducted on personal best achievements and performance regarding the National Championship; however, these data aligned with the qualitatively superior adaptations in VO_2_max seen in the HRV-guided training group and, thus, we found them worthy of consideration.

## 5. Conclusions

Our findings highlight the apparent utility of HRV-guided training to optimize cardiorespiratory benefits and race performance in competitive adolescent runners participating in a short-duration altitude training camp. Future studies to determine the value of HRV-guided training in a larger sample, diverse groups, and under other unique environmental conditions (i.e., heat, cold, etc.) are of interest.

## Figures and Tables

**Figure 1 biosensors-11-00077-f001:**
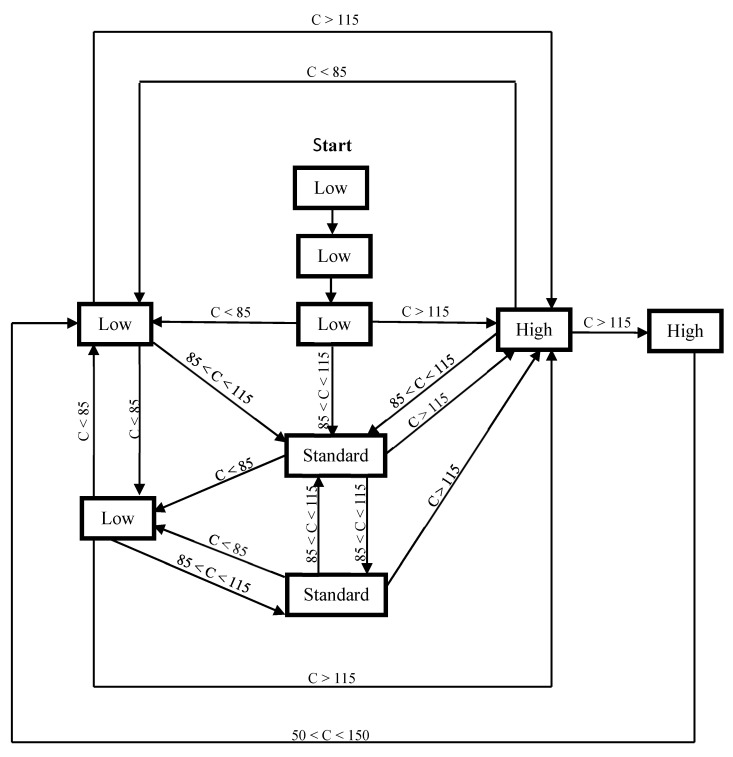
Diagram of the training load used for the heart rate variability (HRV) measuring group. Note: C—compensation.

**Figure 2 biosensors-11-00077-f002:**
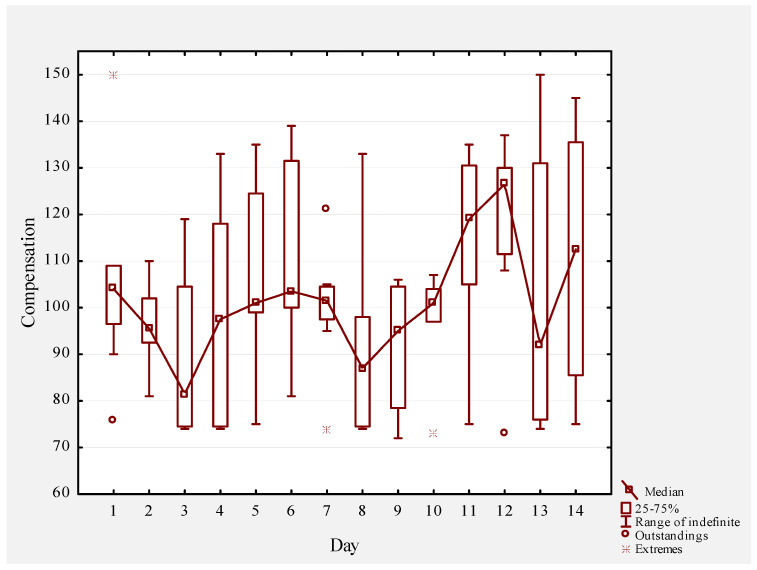
HRV compensation values measured during the 2-week altitude training camp.

**Table 1 biosensors-11-00077-t001:** Training plan for the training camp at higher altitudes.

Day	Morning	Afternoon
Day 1	No training	Slow running
Day 2	Slow running	Running exercises
Day 3	Slow running	Circuit training—light strengthening
Day 4	Race tempo	Walking, hiking, slow running
Day 5	Aerobic endurance	Walking, hiking
Day 6	Speed endurance	Slow running
Day 7	Interval running hills	Walking, hiking, strengthening
Day 8	Slow running, or no training	Swimming or no training
Day 9	Endurance interval training	Slow running, strengthening
Day 10	Aerobic endurance	Walking, hiking, strengthening
Day 11	Speed endurance	Walking, hiking
Day 12	Aerobic endurance	Walking, hiking, strengthening
Day 13	Interval running hills	Walking, hiking
Day 14	Aerobic endurance	No training or walking, hiking

**Table 2 biosensors-11-00077-t002:** VO_2_max values before and after the training camp in the control and HRV-guided training groups.

Group	Measurement Time	VO_2_max (mL·min^−1^·kg^−1^)
Control group	Pre	65.3 ± 3.9
	Post	66.1 ± 3.9
HRV-guided group	Pre	65.6 ± 4.1
	Post	68.4 ± 4.2 *^,#^

* *p* < 0.05 vs. pre-camp measurement in the same group, ^#^
*p* < 0.05 vs. control group at the corresponding time point.

**Table 3 biosensors-11-00077-t003:** Number of athletes in the HRV-guided and control training groups setting personal records or achieving success in the National Championship.

Group	Achieving Success	Personal Best Time	National Championship
Control group	Yes	6	5
	No	2	3
HRV-guided group	Yes	8	6
	No	0	2

## Data Availability

Not applicable.

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
