# Peer review of "Superior Adaptations in Adolescent Runners Using Heart Rate Variability (HRV)-Guided Training at Altitude"

_biosensors, 2021, doi:10.3390/bios11030077_

Round 1

Reviewer 1 Report

The authors investigate the effect of HRV-guided training (compared to standard periodization) in terms of performance and physiological (VO2max) improvements in adolescents training at altitude for 2 weeks

Previous works: the work of Alejandro Javaloyes on HRV-guided training is ignored in the paper, it would be valuable to add it and also to discuss differences as his group showed similar results but with a slightly different approach (similar to the one of Vesterinen, so using the baseline - 7 days moving average - and normal values or SWC - typically 30 days - to determine when to reduce intensity). The altitude stressor obviously challenges this approach, which might motivate your method, using daily scores. Maybe adding a bit on these aspects in the discussion can be beneficial to the reader that is familiar with other protocols.

I find it somewhat strange that work focused on physiological responses (HRV) uses external load to adapt the load to altitude (speed instead of for example heart rate, internal load during exercise). Please clarify this choice.

The major issue of the paper is the lack of transparency in the methods. This must be revised. In particular, the methodology is unclear, this statement is not explaining in any way how the data is used: "values were then used to compute compensation values on a scale of 50-150 (%)" please clarify.

What are you doing with the 3 measurements? I am sure you are aware that HRV can me quantified in different ways, the paper does not mention what HRV feature is used? again: what's HRV compensation? 

Please clarify all of the above in a way that the work can be understood and reproduced by others. 

Reviewer 2 Report

I do not have any comments about the Introduction.

Material and Methods
There is a problem with the design of the study. The authors state the following sentence: "The runners were matched for gender, height, weight, and cardiorespiratory fitness, and allocated to one of two groups: an HRV guided training group and a control group." It is the full description of the dividing of probands into the groups. There is no description of how and why the authors consider the mentioned parameters. However, this sentence mainly leads the reader to the idea that there is no guaranteed impartiality of the results. A standard approach should use randomization of probands into the groups (even in that small number of probands). The hypothesis of the study is enough general for this purpose. Then, there is mentioned the word "randomization" in the discussion in line 207. It is almost the end of the article. Moreover, there is still no more information about randomization. So, the problem remains.

Further, there is a question if the number of probands is sufficient for some general conclusion. The discussion of the paper contains the statement: "Other limitations of this work include our relatively small number of participants...".  However, there is no mention of power analysis in the text. I understand that the number of athletes gives the number of probands, and it is hard to increase this number somehow. However, power analysis can objectively show that the achieved results are sufficient to generalize or not, and it is necessary to provide more experiments.

In Figure 1, the HRV compensation algorithm is very complex. Conversely, the description in the text is very brief. There is a question of how important is the depicted algorithm for the study. Is it part of the results because it is the original training control method due to HRV compensation.  There are no references to other studies in this part of the text, so it seems that the algorithm is original. Therefore, it should be described how the authors determined the compensation thresholds for a change of the training approach.
Finally, the technical quality of figure 1 is not good. 

Authors used the wrong type of test (the two-way ANOVA) for the evaluation of data. The Repeated measure ANOVA should be used in this case, because this test is specifically dedicated to the situation of two groups observed during a defined time. 

Results
There are two presented results. The first is the graph of HRV compensation. There is no clear what the compensation graph provides because there is no more in-depth description of this result. In the discussion, there is mentioned that there is an upward trend in HRV values beginning in day 9 of the training camp. However, the results part do not contain any evaluation of the compensation progress in the 9 days.  
How significant are differences between probands, is it evaluated. The sense of this result is a little bit mysterious.
The main results are in table 2. However, as I mentioned above, there is a problem with the type of used test. Moreover, the mechanism of dividing probands into groups is not clear, so there cannot be excluded bias of the evaluation.
The information about achieving success cannot be taken into account. Not because of the nature of the information, but because of the small number of probands. As it seems, the authors are aware of this fact, and therefore, do not test this result for significance.

In summary, there is not sufficiently described the design and impact (generalization) of the study. There is not an appropriate description and evaluation of the results.
In this form, the article is not eligible to be published in the journal with the impact factor of 3.24.

Reviewer 3 Report

It's will be better if there is a HRV measurement in  control group and the athletes were not informed of the results. In this condition, we can get the degree of subjective bias of each athlete. Moreover, the relationship between compensation values and subjective strain can be explored. The authors should descibe how to calculate the compensation value.

There is only one HRV test a day after waking and bladder voiding, whether it means that the athletes' training intensity in a day is completely determined by the results of this test. Whether this will result in insufficient training intensity or overtraining of athletes?  Whether the real-time monitoring of athletes' HRV can be realized?

Round 2

Reviewer 2 Report

My comments on the authors' responses are in the attached file.

Reviewer 3 Report

In Figure 1, the content of the expression is not intuitive enough, and cannot cover all possible situations that the athletes may experience during training. For example, the athletes have a standard or high compensation value  during training the next day, and there is no start label in the figure. Therefore, Figure 1 needs more improvements.

Figure 2 mainly shows the HRV changes of the experimental group, but the necessary text is missing in the results chapter. Perhaps, the heart rate variability of the control group is also listed to provide visual support for the feasibility of the experiment and the verification of the results.

Because the sample size is too small, it is impossible to achieve fairness in the actual experiment grouping process, so the experimental results  in Table 3 cannot effectively support the experimental hypothesis. So there should be more data evaluation on the individual's training effect.

In summary, there should be more accurate algorithm explanations, and the verification of experimental hypotheses requires more adequate parameter evaluation.
